# On the Consistency Loss for Leveraging Augmented Data to Learn Robust and Invariant Representations

## Abstract

Data augmentation is one of the most popular techniques for improving the robustness of neural networks. In addition to directly training the model with original samples and augmented samples, a torrent of methods regularizing the distance between embeddings/representations of the original samples and their augmented counterparts have been introduced. In this paper, we explore these various regularization choices, seeking to provide a general understanding of how we should regularize the embeddings. Our analysis suggests the ideal choices of regularization correspond to various assumptions. With an invariance test, we argue that regularization is important if the model is to be used in a broader context than accuracy-driven setting because non-regularized approaches are limited in learning the concept of invariance, despite equally high accuracy. Finally, we also show that the generic approach we identified (squared $\ell_2$ norm regularized augmentation) outperforms several recent methods, which are each specially designed for one task and significantly more complicated than ours, over three different tasks.

## 1 Introduction

Recent advances in deep learning has delivered remarkable empirical performance over *i.i.d* test data, and the community continues to investigate the more challenging and realistic scenario when models are tested in robustness over non-*i.i.d* data (*e.g.*, Ben-David et al., 2010; Szegedy et al., 2013). Recent studies suggest that one cause of the fragility is the model's tendency in capturing undesired signals (Wang et al., 2020), thus combating this tendency may be a key to robust models.

To help models ignore the undesired signals, data augmentation (*i.e.*, diluting the undesired signals of training samples by applying transformations to existing examples) is often used. Given its widely usage, we seek to answer the question: *how should we train with augmented samples so that the assistance of augmentation can be taken to the fullest extent to learn robust and invariant models?*

In this paper, We analyze the generalization behaviors of models trained with augmented data and associated regularization techniques. We investigate a set of assumptions and compare the worst-case expected risk over unseen data when *i.i.d* samples are allowed to be transformed according to a function belonging to a family. We bound the expected risk with terms that can be computed during training, so that our analysis can inspire how to regularize the training procedure. While all the derived methods have an upper bound of the expected risk, with progressively stronger assumptions, we have progressively simpler regularization, allowing practical choices to be made according to the understanding of the application. Our contributions of this paper are as follows:

- We offer analyses of the generalization behaviors of augmented models trained with different regularizations: these regularizations require progressively stronger assumptions of the data and the augmentation functions, but progressively less computational efforts. For example, with assumptions pertaining to augmentation transformation functions, the Wasserstein distance over the original and augmented empirical distributions can be calculated through simple $\ell_1$ norm distance.

- We test and compare these methods and offer practical guidance on how to choose regularizations in practice. In short, regularizing the squared $\ell_2$ distance of logits between the augmented samples and original samples is a favorable method, suggested by both theoretical and empirical evidence.

- With an invariance test, we argue that vanilla augmentation does not utilize the augmented samples to the fullest extent, especially in learning invariant representations, thus may not be ideal unless the only goal of augmentation is to improve the accuracy over a specific setting.

## 2 RELATED WORK & KEY DIFFERENCES

Data augmentation has been used effectively for years. Tracing back to the earliest convolutional neural networks, we notice that even the LeNet applied on MNIST dataset has been boosted by mixing the distorted images to the original ones (LeCun et al., 1998). Later, the rapidly growing machine learning community has seen a proliferate development of data augmentation techniques (*e.g.*, flipping, rotation, blurring *etc.*) that have helped models climb the ladder of the state-of-the-art (one may refer to relevant survey (Shorten & Khoshgoftaar, 2019) for details). Recent advances expanded the conventional concept of data augmentation and invented several new approaches, such as leveraging the information in unlabelled data (Xie et al., 2019), automatically learning augmentation functions (Ho et al., 2019; Hu et al., 2019; Wang et al., 2019c; Zhang et al., 2020; Zoph et al., 2019), and generating the samples (with constraint) that maximize the training loss along training (Fawzi et al., 2016), which is later widely accepted as adversarial training (Madry et al., 2018).

While the above works mainly discuss how to generate the augmented samples, in this paper, we mainly answer the question about how to train the models with augmented samples. For example, instead of directly mixing augmented samples with the original samples, one can consider regularizing the representations (or outputs) of original samples and augmented samples to be close under a distance metric (also known as a consistency loss). Many concrete ideas have been explored in different contexts. For example, $\ell_2$ distance and cosine similarities between internal representations in speech recognition (Liang et al., 2018), squared $\ell_2$ distance between logits (Kannan et al., 2018), or KL divergence between softmax outputs (Zhang et al., 2019a) in adversarially robust vision models, Jensen–Shannon divergence (of three distributions) between embeddings for texture invariant image classification (Hendrycks et al., 2020). These are but a few highlights of the concrete and successful implementations for different applications out of a huge collection (*e.g.*, (Wu et al., 2019; Guo et al., 2019; Zhang et al., 2019b; Shah et al., 2019; Asai & Hajishirzi, 2020; Sajjadi et al., 2016; Zheng et al., 2016; Xie et al., 2015)), and one can easily imagine methods permuting these three elements (distance metrics, representation or outputs, and applications) to be invented. Even further, although we are not aware of the following methods in the context of data augmentation, given the popularity of GAN (Goodfellow, 2016) and domain adversarial neural network (Ganin et al., 2016), one can also expect the distance metric generalizes to a specialized discriminator (*i.e.* a classifier), which can be intuitively understood as a calculated (usually maximized) distance measure, Wasserstein-1 metric as an example (Arjovsky et al., 2017; Gulrajani et al., 2017).

**Key Differences:** With this rich collection of regularizing choices, which one method should we consider in general? More importantly, do we actually need the regularization at all? These questions are important for multiple reasons, especially considering that there are paper suggesting that these regularizations may lead to worse results (Jeong et al., 2019). In this paper, we answer the first question with a proved upper bound of the worst case generalization error, and our upper bound explicitly describes what regularizations are needed. For the second question, we will show that regularizations can help the model to learn the concept of invariance.

There are also several previous discussions regarding the detailed understandings of data augmentation (Yang et al., 2019; Chen et al., 2019; Hernández-García & König, 2018; Rajput et al., 2019; Dao et al., 2019), among which, (Yang et al., 2019) is probably the most relevant as it also defends the usage of regularizations. However, we believe our discussions are more comprehensive and supported theoretically, since our analysis directly suggests the ideal regularization. Also, empirically, we design an invariance test in addition to the worst-case accuracy used in the preceding work.

## 3 TRAINING STRATEGIES WITH AUGMENTED DATA

**Notations** $(\mathbf{X}, \mathbf{Y})$ denotes the data, where $\mathbf{X} \in \mathcal{R}^{n \times p}$ and $\mathbf{Y} \in \{0, 1\}^{n \times k}$ (one-hot vectors for $k$ classes), and $f(\cdot, \theta)$ denotes the model, which takes in the data and outputs the softmax (probabilities of the prediction) and $\theta$ denotes the corresponding parameters. $g()$ completes the prediction (*i.e.*, mapping softmax to one-hot prediction). $l(\cdot, \cdot)$ denotes a generic loss function. $a(\cdot)$ denotes a

transformation that alters the undesired signals of a sample, *i.e.*, the data augmentation method. $a \in \mathcal{A}$, which is the set of transformation functions. $\mathcal{P}$ denotes the distribution of $(\mathbf{x}, \mathbf{y})$. For any sampled $(\mathbf{x}, \mathbf{y})$, we can have $(a(\mathbf{x}), \mathbf{y})$, and we use $\mathcal{P}_a$ to denote the distribution of these transformed samples. $r(\cdot; \theta)$ denotes the risk of model $\theta$. $\widehat{\cdot}$ denotes the estimated term $\cdot$.

## 3.1 WELL-BEHAVED DATA TRANSFORMATION FUNCTION

Despite the strong empirical performance data augmentation has demonstrated, it should be intuitively expected that the performance can only be improved when the augmentation is chosen wisely. Therefore, before we proceed to analyze the behaviors of training with data augmentations, we need first regulate some basic properties of the data transformation functions used. Intuitively, we will consider the following three properties.

- "Dependence-preservation" with two perspectives: Label-wise, the transformation cannot alter the label of the data, which is a central requirement of almost all the data augmentation practice. Feature-wise, the transformation will not introduce new dependencies between the samples.
- "Efficiency": the augmentation should only generate new samples of the same label as minor perturbations of the original one. If a transformation violates this property, there should exist other simpler transformations that can generate the same target sample.
- "Vertices": There are extreme cases of the transformations. For example, if one needs the model to be invariant to rotations from $0°$ to $60°$, we consider the vertices to be $0°$ rotation function (thus identity map) and $60°$ rotation function. In practice, one usually selects the transformation vertices with intuitions and domain knowledge.

We now formally define these three properties. The definition will depend on the model, thus these properties are not only regulating the transformation functions, but also the model. We introduce the Assumptions A1-A3 corresponding to the properties.

**A1**: Dependence-preservation: the transformation function will not alter the dependency regarding the label (*i.e.*, for any $a() \in \mathcal{A}$, $a(\mathbf{x})$ will have the same label as $\mathbf{x}$) or the features (*i.e.*, for any $a_1(), a_2() \in \mathcal{A}$, $a_1(\mathbf{x}_1) \perp\!\!\!\perp a_1(\mathbf{x}_2)$ for any $\mathbf{x}_1, \mathbf{x}_2 \in \mathbf{X}$ that $\mathbf{x}_1 \neq \mathbf{x}_2$).

**A2**: Efficiency: for $\widehat{\theta}$ and any $a() \in \mathcal{A}$, $f(a(\mathbf{x}); \widehat{\theta})$ is closer to $\mathbf{x}$ than any other samples under a distance metric $d_e(\cdot, \cdot)$, *i.e.*, $d_e(f(a(\mathbf{x}); \widehat{\theta}), f(\mathbf{x}; \widehat{\theta})) \leq \min_{\mathbf{x}' \in \mathbf{X}_{-\mathbf{x}}} d_e(f(a(\mathbf{x}); \widehat{\theta}), f(\mathbf{x}'; \widehat{\theta}))$.

**A3**: Vertices: For a model $\widehat{\theta}$ and a transformation $a()$, we use $\mathcal{P}_{a,\widehat{\theta}}$ to denote the distribution of $f(a(\mathbf{x}); \widehat{\theta})$ for $(\mathbf{x}, \mathbf{y}) \sim \mathcal{P}$. "Vertices" argues that exists two extreme elements in $\mathcal{A}$, namely $a^+$ and $a^-$, with certain metric $d_x(\cdot, \cdot)$, we have

$$d_x(\mathcal{P}_{a^+,\widehat{\theta}}, \mathcal{P}_{a^-,\widehat{\theta}}) = \sup_{a_1, a_2 \in \mathcal{A}} d_x(\mathcal{P}_{a_1,\widehat{\theta}}, \mathcal{P}_{a_2,\widehat{\theta}}) \tag{1}$$

Note that $d_x(\cdot, \cdot)$ is a metric over two distributions and $d_e(\cdot, \cdot)$ is a metric over two samples. Also, slightly different from the intuitive understanding of "vertices" above, **A3** regulates the behavior of embedding instead of raw data. All of our follow-up analysis will require **A1** to hold, but with more assumptions held, we can get computationally lighter methods with bounded error.

## 3.2 BACKGROUND, ROBUSTNESS, AND INVARIANCE

One central goal of machine learning is to understand the generalization error. When the test data and train data are from the same distribution, many previous analyses can be sketched as:

$$r_{\mathcal{P}}(\widehat{\theta}) \leq \widehat{r}_{\mathcal{P}}(\widehat{\theta}) + \phi(|\Theta|, n, \delta) \tag{2}$$

which states that the expected risk can be bounded by the empirical risk and a function of hypothesis space $|\Theta|$ and number of samples $n$; $\delta$ accounts for the probability when the bound holds. $\phi()$ is a function of these three terms. Dependent on the details of different analyses, different concrete examples of this generic term will need different assumptions. We use a generic assumption **A4** to denote the assumptions required for each example. More concrete discussions are in Appendix A

**Robustness** In addition to the generalization error above, we also study the robustness by following the established definition as in the worst case expected risk when the test data is allowed to be shifted to some other distributions by transformation functions in $\mathcal{A}$. Formally, we study

$$r_{\mathcal{P}'}(\widehat{\theta}) = \mathbb{E}_{(\mathbf{x},\mathbf{y}) \sim \mathcal{P}} \max_{a \sim \mathcal{A}} \mathbb{I}(g(f(a(\mathbf{x}); \widehat{\theta})) \neq \mathbf{y}) \tag{3}$$

As $r_{\mathcal{P}}(\widehat{\theta}) \leq r_{\mathcal{P}'}(\widehat{\theta})$, we only need to study (3). We will analyze (3) in different scenarios involving different assumptions and offer formalizations of the generalization bounds under each scenario. Our bounds shall also immediately inspire the development of methods in each scenario as the terms involved in our bound are all computable within reasonable computational loads.

**Invariance** In addition to robustness, we are also interested in whether the model learns to be invariant to the undesired signals. Intuitively, if data augmentation is used to help dilute the undesired signals from data by altering the undesired signals with $a() \in \mathcal{A}$, a successfully trained model with augmented data will map the raw data with various undesired signals to the same embedding. Thus, we study the following metric to quantify the model's ability in learning invariant representations:

$$I(\widehat{\theta}, \mathcal{P}) = \sup_{a_1, a_2 \in \mathcal{A}} d_x(\mathcal{P}_{a_1, \widehat{\theta}}, \mathcal{P}_{a_2, \widehat{\theta}}), \tag{4}$$

where $\mathcal{P}_{a, \widehat{\theta}}$ to denote the distribution of $f(a(\mathbf{x}); \widehat{\theta})$ for $(\mathbf{x}, \mathbf{y}) \sim \mathcal{P}$. $d_x()$ is a distance over two distributions, and we suggest to use Wasserstein metric given its favorable properties (*e.g.*, see practical examples in Figure 1 of (Cuturi & Doucet, 2014) or theoretical discussions in (Villani, 2008)). Due to the difficulties in assessing $f(a(\mathbf{x}); \widehat{\theta})$ (as it depends on $\widehat{\theta}$), we mainly study (4) empirically, and argue that models trained with explicit regularization of the empirical counterpart of (4) will have favorable invariance property.

### 3.3 Worst-case Augmentation (Adversarial Training)

We consider robustness first. (3) can be written equivalently into the expected risk over a pseudo distribution $\mathcal{P}'$ (see Lemma 1 in (Tu et al., 2019)), which is the distribution that can sample the data leading to the worst expected risk. Thus, equivalently, we can consider $\sup_{\mathcal{P}' \in T(\mathcal{P}, \mathcal{A})} r_{\mathcal{P}'}(\widehat{\theta})$. With an assumption relating the worst distribution of expected risk and the worst distribution of the empirical risk (namely, **A5**, in Appendix A), the bound of our interest (*i.e.*, $\sup_{\mathcal{P}' \in T(\mathcal{P}, \mathcal{A})} r_{\mathcal{P}'}(\widehat{\theta})$) can be analogously analyzed through $\sup_{\mathcal{P}' \in T(\mathcal{P}, \mathcal{A})} \widehat{r}_{\mathcal{P}'}(\widehat{\theta})$. By the definition of $\mathcal{P}'$, we can have:

**Lemma 3.1.** *With Assumptions A1, A4, and A5, with probability at least $1 - \delta$, we have*

$$\sup_{\mathcal{P}' \in T(\mathcal{P}, \mathcal{A})} r_{\mathcal{P}'}(\widehat{\theta}) \leq \frac{1}{n} \sum_{(\mathbf{x}, \mathbf{y}) \sim \mathcal{P}} \sup_{a \in \mathcal{A}} \mathbb{I}(g(f(a(\mathbf{x}); \widehat{\theta})) \neq \mathbf{y}) + \phi(|\Theta|, n, \delta) \tag{5}$$

This result is a straightforward follow-up of the preceding discussions. In practice, it aligns with the adversarial training (Madry et al., 2018), a method that has demonstrated impressive empirical successes in the robust machine learning community.

While the adversarial training has been valued by its empirical superiorities, it may still have the following two directions that can be improved: firstly, it lacks an explicit enforcement of the concept of invariance between the original sample and the transformed sample; secondly, it assumes that elements of $\mathcal{A}$ are enumerable, thus $\frac{1}{n} \sum_{(\mathbf{x}, \mathbf{y}) \sim \mathcal{P}} \sup_{a \in \mathcal{A}} \mathbb{I}(g(f(a(\mathbf{x}); \widehat{\theta})) \neq \mathbf{y})$ is computable. The remaining discussions expand along these two directions.

### 3.4 Regularized Worst-case Augmentation

To force the concept of invariance, the immediate solution might be to apply some regularizations to minimize the distance between the embeddings learned from the original sample and the ones learned from the transformed samples. We have offered a summary of these methods in Section 2.

To have a model with small invariance score, the direct approach will be regularizing the empirical counterpart of (4). We notice that existing methods barely consider this regularization, probably because of the computational difficulty of Wasserstein distance. Conveniently, we have the following result that links the $\ell_1$ regularization to the Wasserstein-1 metric in the context of data augmentation.

**Proposition 3.2.** *With A2, and $d_e(\cdot, \cdot)$ in A2 chosen to be $\ell_1$ norm, for any $a \in \mathcal{A}$, we have*

$$\sum_i ||f(\mathbf{x}_i; \widehat{\theta}) - f(a(\mathbf{x}_i); \widehat{\theta})||_1 = W_1(f(\mathbf{x}; \widehat{\theta}), f(a(\mathbf{x}); \widehat{\theta})) \tag{6}$$

This result conveniently allows us to use $\ell_1$ norm distance to replace Wasserstein metric, integrating the advantages of Wasserstein metric while avoiding practical issues such as computational complexity and difficulty to pass the gradient back during backpropagation.

We continue to discuss the generalization behaviors. Our analysis remains in the scope of multi-class classification, where the risk is evaluated as misclassification rate, and the model is optimized with cross-entropy loss (with the base chosen to be log base in cross-entropy loss). This setup aligns with **A4**, and should represent the modern neural network studies well enough.

Before we proceed, we need another technical assumption **A6** (details in Appendix A), which can be intuitively considered as a tool that allows us to relax classification error into cross-entropy error, so that we can bound the generalization error with the terms we can directly optimize during training.

We can now offer another technical result:

**Theorem 3.3.** *With Assumptions A1, A2, A4, A5, and A6, and $d_e(\cdot, \cdot)$ in A2 is $\ell_1$ norm, with probability at least $1 - \delta$, the worst case generalization risk will be bounded as*

$$\sup_{\mathcal{P}' \in T(\mathcal{P}, \mathcal{A})} r_{\mathcal{P}'}(\widehat{\theta}) \leq \widehat{r}_{\mathcal{P}}(\widehat{\theta}) + \sum_i ||f(\mathbf{x}_i; \widehat{\theta}) - f(\mathbf{x}'_i; \widehat{\theta})||_1 + \phi(|\Theta|, n, \delta) \tag{7}$$

*and $\mathbf{x}' = a(\mathbf{x})$, where $a = \arg\max_{a \in \mathcal{A}} \mathbf{y}^\top f(a(\mathbf{x}); \widehat{\theta})$.*

This technical result also immediately inspires the method to guarantee worst case performance, as well as to explicitly enforce the concept of invariance. Notice that $a = \arg\max_{a \in \mathcal{A}} \mathbf{y}^\top f(a(\mathbf{x}); \widehat{\theta})$ is simply selecting the augmentation function maximizing the cross-entropy loss, a standard used by many worst case augmenting method (*e.g.*, Madry et al., 2018).

## 3.5 Regularized Training with Vertices

As $\mathcal{A}$ in practice is usually a set with a large number of (and possibly infinite) elements, we may not always be able to identify the worst case transformation function with reasonable computational efforts. This limitation also prevents us from effective estimating the generalization error as the bound requires the identification of the worst case transformation.

Our final discussion is to leverage the vertex property of the transformation function to bound the worst case generalization error:

**Lemma 3.4.** *With Assumptions A1-A6, and $d_e(\cdot, \cdot)$ in A2 chosen as $\ell_1$ norm distance, $d_x(\cdot, \cdot)$ in A3 chosen as Wasserstein-1 metric, assuming there is a $a'() \in \mathcal{A}$ where $\widehat{r}_{\mathcal{P}_{a'}}(\widehat{\theta}) = \frac{1}{2}\big(\widehat{r}_{\mathcal{P}_{a+}}(\widehat{\theta}) + \widehat{r}_{\mathcal{P}_{a-}}(\widehat{\theta})\big)$, with probability at least $1 - \delta$, we have:*

$$\sup_{\mathcal{P}' \in T(\mathcal{P}, \mathcal{A})} r_{\mathcal{P}'}(\widehat{\theta}) \leq \frac{1}{2}\big(\widehat{r}_{\mathcal{P}_{a+}}(\widehat{\theta}) + \widehat{r}_{\mathcal{P}_{a-}}(\widehat{\theta})\big) + \sum_i ||f(a^+(\mathbf{x}_i); \widehat{\theta}) - f(a^-(\mathbf{x}'); \widehat{\theta})||_1 + \phi(|\Theta|, n, \delta)$$

This result inspires the method that can directly guarantee the worst case generalization result and can be optimized conveniently without searching for the worst-case transformations. However, this method requires a good domain knowledge of the vertices of the transformation functions.

## 3.6 Engineering Specification of Relevant Methods

Our theoretical analysis has lead to a line of methods, however, not every method can be effectively implemented, especially due to the difficulties of passing gradient back for optimizations. Therefore, to boost the influence of the loss function through backpropagation, we recommend to adapt the methods with the following two changes: 1) the regularization is enforced on logits instead of softmax; 2) we use squared $\ell_2$ norm instead of $\ell_1$ norm because $\ell_1$ norm is not differentiable everywhere. We discuss the effects of these compromises in ablation studies in Appendix E.

Also, in the cases where we need to identify the worst case transformation functions, we iterate through all the transformation functions and identify the function with the maximum loss.

Overall, our analysis leads to the following main training strategies:

- VA (vanilla augmentation): mix the augmented samples of a vertex function to the original ones for training (original samples are considered as from another vertex in following experiments).
- VWA (vanilla worst-case augmentation): at each iteration, identify the worst-case transformation functions and train with samples generated by them (also known as adversarial training).
- RA (regularized augmentation): regularizing the squared $\ell_2$ distance over logits between the original samples and the augmented samples of a fixed vertex transformation function.
- RWA (regularized worst-case augmentation): regularizing the squared $\ell_2$ distance over logits between the original samples and the worst-case augmented samples identified at each iteration.

## 4 EXPERIMENTS

We first use some synthetic experiments to verify our assumptions and inspect the consequences when the assumptions are not met (in Appendix C). Then, in the following paragraphs, we test the methods discussed to support our arguments in learning robustness and invariance. Finally, we show the power of our discussions by competing with advanced methods designed for specific tasks.

### 4.1 EXPERIMENTS FOR LEARNING ROBUST & INVARIANT REPRESENTATION

**Experiment Setup:** We first test our arguments with two data sets and three different sets of the augmentations. We study MNIST dataset with LeNet architecture, and CIFAR10 dataset with ResNet18 architecture. To examine the effects of the augmentation strategies, we disable all the heuristics that are frequently used to boost the test accuracy of models, such as the default augmentation many models trained for CIFAR10 adopted, and the BatchNorm (also due to the recent arguments against the effects of BatchNorm in learning robust features (Wang et al., 2020)), although forgoing these heuristics will result in a lower overall performance than one usually expects.

We consider three different sets of transformation functions: texture, rotation, and contrast. The details of these transformation functions and the experiment setup are in Appendix D.

We consider three different evaluation metrics:

- Clean: test accuracy on the original test data, mainly reported as a reference for other metrics.
- Robustness: the worst accuracy when each sample can be transformed with $a \in \mathcal{A}$.
- Invariance: A metric to test whether the models learns the concept of invariance (details to follow).

**Invariance-test:** To test whether a model can truly learns the concept of invariance within $\mathcal{A} = \{a_1(), a_2(), \ldots, a_t()\}$ of $t$ elements, we design a new evaluation metric: for a sampled collection of data of the sample label $i$, denoted as $\mathbf{X}^{(i)}$, we generate the transformed copies of it with $\mathcal{A}$, resulting in $\mathbf{X}_{a_1}^{(i)}, \mathbf{X}_{a_2}^{(i)}, \ldots, \mathbf{X}_{a_t}^{(i)}$. We combined these copies into a dataset, denoted as $\mathcal{X}^{(i)}$. For every sample $\mathbf{x}$ in $\mathcal{X}^{(i)}$, we retrieve its $t$ nearest neighbors of other samples in $\mathcal{X}^{(i)}$, and calculate the overlap of the retrieved samples and $\{a_1(\mathbf{x}), a_2(\mathbf{x}), \ldots, a_t(\mathbf{x})\}$. Since the identify map is in $\mathcal{A}$, so the calculated overlap score will be in $[1/t, 1]$. The distance used is $d(\cdot, \cdot) = ||f(\cdot; \widehat{\theta}) - f(\cdot; \widehat{\theta})||_1$, where $\widehat{\theta}$ is the model we are interested to examine. Finally, we report the averaged score for every label. Thus, a high overlap score indicates the prediction of model $\widehat{\theta}$ is invariant to the augmentation functions in $\mathcal{A}$. If we use other distance functions, the reported values may differ, but we notice that the rank of the methods compared in terms of this test barely changes.

**Results:** We show the results in Table 1 and Table 6 (in Appendix) for MNIST and CIFAR10 respectively. Table 1 shows that RWA is generally a superior method, in terms of all the metrics, especially the invariance evaluation as it shows a much higher invariance score than competing methods. We believe this advantage of invariance comes from two sources: regularizations and the fact that RWA has seen all the augmentation functions in $\mathcal{A}$. In comparison, RA also has regularization but only

| | Texture | | | Rotation | | | Contrast | | |
|---|---|---|---|---|---|---|---|---|---|
| | C | R | I | C | R | I | C | R | I |
| Base | 0.9921 | 0.9860 | 0.9236 | 0.9921 | 0.2960 | 0.2056 | 0.9921 | 0.2699 | 0.2003 |
| VA | **0.9928** | 0.9906 | 0.9876 | 0.9884 | 0.9336 | 0.5628 | 0.9922 | 0.9837 | 0.4153 |
| RA | 0.9909 | 0.9904 | **1** | 0.9930 | 0.9525 | 0.6044 | 0.9936 | 0.9823 | 0.4166 |
| VWA | 0.9922 | 0.9903 | 0.9940 | 0.9466 | 0.9408 | 0.6284 | 0.536 | 0.4470 | 0.2210 |
| RWA | 0.9915 | **0.9911** | **1** | **0.9934** | **0.9882** | **0.8856** | **0.994** | **0.9893** | **0.8786** |

Table 1: Results of MNIST data ("C" stands for clean accuracy, "R" stands for robustness, and "I" stands for invariance score): invariance score shows big differences while accuracy does not.

sees the vertices in $\mathcal{A}$, so the invariance score of RA is not compatitable to RWA, although better than VA. Table 6 roughly tells the same story. More discussions are in Appendix D.

**Other results (Appendix E):** The strength of RWA can also be shown in several other different scenarios, even in the out-of-domain test scenario where the transformation functions are not in $\mathcal{A}$. RWA generally performs the best, although not the best in every single test. We also perform ablation test to validate the choice of squared $\ell_2$ norm over logits in contrast to other distance metrics. Our choice performs the best in the worst-case performance. This advantage is expected as our choice is validated by theoretical arguments as well as consideration of engineering convenience.

Overall, the empirical performances align with our expectation from the theoretical discussion: while all methods discussed have a bounded worst case performance, we do not intend to compare the upper bounds because smaller upper bounds do not necessarily guarantee a smaller risk. However, worst case augmentation methods tend to show a better worst case performances because they have been augmented with all the elements in $\mathcal{A}$. Also, there is no clear evidence suggesting the difference between augmentation methods and its regularized versions in terms of the worst case performance, but it is clear that regularization helps to learn the concept of invariance.

## 4.2 Comparison to Advanced Methods

Finally, we also compete our generic data augmentation methods against several specifically designed methods in different applications. We use the four generic methods (VA, RA, VWA, and RWA) with generic transformation functions ($\mathcal{A}$ of "rotation", "contrast", or "texture" used in the synthetic experiments). We compare our methods with techniques invented for three different topics of study (rotation invariant, texture perturbation, and cross-domain generalization), and each of these topics has seen a long line of method development. We follow each own tradition (*e.g.*, rotation methods are usually tested in CIFAR10 dataset, seemingly due to the methods' computational requirements), test over each own most challenging dataset (*e.g.*, ImageNet-Sketch is the most recent and challenging dataset in domain generalization, although less studied), and report each own evaluation metric (*e.g.*, methods tested with ImageNet-C are usually evaluated with mCE).

Overall, the performances of our generic methods outperform these advanced SOTA techniques. Thus, the main conclusion, as validated by these challenging scenarios, are (1) usage of data augmentation can outperform carefully designed methods; (2) usage of the consistency loss can further improve the performances; (3) regularized worst-case augmentation generally works the best.

Due to the limitation of space, we leave the background details of these experiments in Appendix F, where we introduce the detailed experiment settings, and explain the acronyms in Tables 2-4.

**Rotation-invariant Image Classification** We test the models with nine different rotations including $0°$. Augmentation related methods only use the $\mathcal{A}$ of "rotation" in synthetic experiments, so the testing scenario goes beyond what the augmentation methods have seen during training. The results in Table 2 strongly endorses the efficacy of augmentation-based methods. Interestingly, regularized augmentation methods probably with the benefit of learning the concept of invariance, tend to behave well in the transformations not considered during training. Also, RA outperforms VWA on average.

**Texture-perturbed ImageNet classification** We also test the performance on the image classification over multiple perturbations. We train the model over standard ImageNet training set and test the model with ImageNet-C data (Hendrycks & Dietterich, 2019), which is a perturbed version of ImageNet by corrupting the original ImageNet validation set with a collection of noises. The results

| | 300 | 315 | 330 | 345 | 0 | 15 | 30 | 45 | 60 | avg. |
|------|--------|--------|--------|--------|--------|--------|--------|--------|--------|--------|
| Base | 0.2196 | 0.2573 | 0.3873 | 0.6502 | 0.8360 | 0.6938 | 0.4557 | 0.3281 | 0.2578 | 0.4539 |
| ST | 0.2391 | 0.2748 | 0.4214 | 0.7049 | 0.8251 | 0.7147 | 0.4398 | 0.2838 | 0.2300 | 0.4593 |
| GC | 0.1540 | 0.1891 | 0.2460 | 0.3919 | 0.5859 | 0.4145 | 0.2534 | 0.1827 | 0.1507 | 0.2853 |
| ETN | 0.3855 | **0.4844** | **0.6324** | **0.7576** | 0.8276 | 0.7730 | 0.7324 | 0.6245 | 0.5060 | 0.6358 |
| VA | 0.2233 | 0.2832 | 0.4318 | 0.6364 | 0.8124 | 0.6926 | 0.5973 | 0.7152 | 0.7923 | 0.5761 |
| RA | 0.3198 | 0.3901 | 0.5489 | 0.7170 | 0.8487 | 0.7904 | 0.7455 | 0.8005 | 0.8282 | 0.6655 |
| VWA | 0.3383 | 0.3484 | 0.3835 | 0.4569 | 0.7474 | 0.866 | 0.8776 | 0.8738 | 0.8629 | 0.6394 |
| RWA | **0.4012** | 0.4251 | 0.4852 | 0.6765 | **0.8708** | **0.8871** | **0.8869** | **0.8870** | **0.8818** | **0.7113** |

Table 2: Comparison to advanced rotation-invariant models. We report the test accuracy on the test sets clockwise rotated, 0°-60° and 300°-360°. Average accuracy is also reported. Augmentation methods only consider 0°-60° clockwise rotations during training.

| | Base | SU | AA | MBP | SIN | AM | AMS | VA | RA | VWA | RWA |
|-------|------|------|------|------|------|--------|------|------|------|------|--------|
| Clean | 23.9 | 24.5 | 22.8 | 23 | 27.2 | **22.4** | 25.2 | 23.7 | 23.6 | 23.3 | **22.4** |
| mCE | 80.6 | 74.3 | 72.7 | 73.4 | 73.3 | 68.4 | 64.9 | 76.3 | 75.6 | 74.8 | **64.6** |

Table 3: Summary comparison to advanced models over ImageNet-C data. Performance reported (mCE) follows the standard in ImageNet-C data: clean error and mCE are both the smaller the better.

| | Base | InfoDrop | HEX | PAR | VA | RA | VWA | RWA |
|-------|--------|----------|--------|--------|--------|--------|--------|--------|
| Top-1 | 0.1204 | 0.1224 | 0.1292 | 0.1306 | 0.1362 | 0.1405 | 0.1432 | **0.1486** |
| Top-5 | 0.2408 | 0.256 | 0.2564 | 0.2627 | 0.2715 | 0.2793 | 0.2846 | **0.2933** |

Table 4: Comparison to advanced cross-domain image classification models, over ImageNet-Sketch dataset. We report top-1 and top-5 accuracy following standards on ImageNet related experiments.

are reported in Table 3, which shows that our generic method can outperform the current SOTA methods after a continued finetuning process with reducing learning rates.

**Cross-domain ImageNet-Sketch Classification** We also compare to the methods used for cross-domain evaluation. with the most challenging setup in this scenario: train the models with standard ImageNet training data, and test the model over ImageNet-Sketch data (Wang et al., 2019a), which is a collection of sketches following the structure ImageNet validation set. Similarly, we only augment the samples with a generic augmentation set ($\mathcal{A}$ of "contrast" in synthetic experiments, Appendix D). The results in Table 4 again support the strength of the correct usage of data augmentation.

## 5 CONCLUSION

In this paper, we conducted a systematic inspection to study the proper regularization techniques that are provably related to the generalization error of a machine learning model, when the test distribution are allowed to be perturbed by a family of transformation functions. With progressively more specific assumptions, we identified progressively simpler methods that can bound the worst case risk. We summarize the main **take-home messages** below:

- Regularizing a norm distance between the logits of the originals samples and the logits of the augmented samples enjoys several merits: the trained model tend to have good worst cast performance, and can learn the concept of invariance (as shown in our invariance test). Although our theory suggests $\ell_1$ norm, but we recommend squared $\ell_2$ norm in practice considering the difficulties of passing the (sub)gradient of $\ell_1$ norm in backpropagation.

- With the vertex assumption held (it usually requires domain knowledge to choose the vertex functions), one can use "regularized training with vertices" method and get good empirical performance in both accuracy and invariance, and the method is at the same complexity order of vanilla training without data augmentation. When we do not have the domain knowledge (thus are not confident in the vertex assumption), we recommend "regularized worst-case augmentation", which has the best performance overall, but requires extra computations to identify the worst-case augmented samples at each iteration.

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
