# OpenReview forum: "On the Consistency Loss for Leveraging Augmented Data to Learn Robust and Invariant Representations"
_ICLR.cc/2021/Conference — Reject_

### Official Review · AnonReviewer2 · 2020-10-27
**Promising theoretical analysis, but there is still unexplained gap between the theoretical and empirical analysis**

**Rating:** 6
**Confidence:** 3

**Review:**

In order to improve the robustness of the learned models, prior work has proposed various data augmentation techniques and different ways of incorporating them into training. This work seeks to provide a general understanding of how we should train with augmented samples in order to learn robust and invariant models from both theoretical and empirical perspectives. More importantly, the authors showed that the regularization of the augmented samples in the training procedure can be inspired from the theoretical analysis since it directly suggests the ideal regularization.

Originality and significance:
The proved upper bound of the worst case generalization error and the suggestion of the ideal regularization is new as far as I know, despite it also leads to the regularization using the worst-case augmentation to introduce invariance as in (Yang et al., 2019).
One issue, though, is that the set of possible data shifts is known beforehand and the best-performing RWA method must see all the augmentation functions. It immediately raises the question of how the method generalizes to unseen augmentations. For example, training with texture and rotation augmentations but testing with the contrast augmentation. This also happens in Section 4.2 when competing with advanced methods. In each experiment, the "ground-truth" augmentation is considered even for cross-domain classification.
Also, the "Contrast" results of VMA in Table 1 look strange. The vanilla augmentation already works very well in the robustness test with a score of 0.9837, but the regularized version only gives a score of 0.4470. Why does this happen?

Clarity:
It is not easy to follow, especially Section 3. There are in total 6 assumptions used in this work, but the authors decided to spend significant space on half of them and put the rest in the supplementary material without any explanation in the main text. Also, it might be better to give an informal illustration of the theoretical results before giving the main theorems so that it would be more likely to be appreciated by a larger community.
On the other hand, there is a gap between the theoretical analysis and the empirical verification. As the authors pointed out, the theoretical analysis advocates the use of the l1 distance while the authors recommend to use the squared l2 norm due to "l1 norm is not differentiable everywhere" and better quantitative results. This raise several questions: i) is there a everywhere differentiable approximation to l1 norm which is better than the squared l2 norm, and ii) what distribution divergence does the squared l2 norm correspond to. Since the empirical analysis in this work in mainly driven by the theoretical work, some discussion regarding the use of squared l2 norm would make the work stronger, especially when "the ideal regularization" does not work well.
Some minor issues: i) I think the symbols used in the last sentence in A1 would be difficult to understand for most of the community, therefore it might be better to explain it in word; ii) I think it should be "argmin" instead of "argmax" in Theorem 3.3. We are aiming for the worst augmentation, but "argmax" would find the augmentation which gives the "best" result; iii) it is almost impossible to know the which competing methods are used in the experiment section without referring to the appendix. Even adding the reference to the methods in each table would help a lot.

---

> ### Author Response · Authors · 2020-11-17
> **Discussion with AnonReviewer2**
>
> Dear AnonReviewer2,
>
> Thank you for the overall positive evaluation of the paper, we appreciate your recognition of the significance of our work with the aim to offer a unified solution to various methods, as well as the theoretical motivation of our work. We are also grateful for the constructive comments raised, although some of them may be an unfortunate consequence of miscommunications. Below, please find our responses to the points raised, as well as our attempt to clarify some points that our paper may have miscommunicated.
>
> * We understand that our paper may have delivered the idea that the ground-truth augmentation strategies are available during the comparison to advanced methods, but they are not available. For examples:
>     * Table 2 reports methods trained only considering 0-60 degrees rotations, tested with 0-60, 300-360 degrees rotations.
>     * Tables 3 & 4 report methods trained with our predefined transformation scopes, while the test data are generated independently, with no direct (mathematical) connections to our transformation scopes.
>     * The "Beyond" columns of Tables 7-12 (six tables in total) are testing the cases when samples are generated with other augmentations within the same augmentation scope.
>     * In the case about training with rotation and testing with texture: please notice that Table 3 & 4 are this in case (the test data of Table 3 & 4 are not rotation, texture, or contrast), and please refer to our next point.
>
> * Why we believe a predefined transformation scope is necessary (even for the human)?
>     * For our discussion to go smoothly, please imagine a friend of us traveled back from Mars and brought back two creatures with arbitrary appearances (e.g., one with a dog ear and cat paw, the other with a cat ear and dog paw), she named them a cat and a dog, and asked us to tell which is which. Without a rough understanding of her naming preferences, there is no way to guarantee the correct classification.
>     * For the same reason, we do not think there can exist any generalization guarantees if we do not have a predefined transformation scope. In fact, even within the same transformation scope, without further assumptions, the generalization results over transformations not considered during training can be intricate to anticipate. e.g., "Beyond" columns of Tables 7-12 do not lead us to a clear conclusion.
>
> * About the unexpected performance of the "Contrast" results of VMA in Table 1. This is because, in the contrast case, the worst-case augmentation strategy identified by the model along training tend to be cases that are generically more difficult to classify. As a result, $\mathbf{x}/4$ and $(1-\mathbf{x})/4$ are fed into the model more frequently than $\mathbf{x}/2$ and $(1-\mathbf{x})/2$, simply because the digits are harder to classify when all the pixels are divided by 4. We had some discussions on this point and experiments to support this conjecture in the last paragraph on Page 18 (appendix). We agree that we need to emphasize this discussion more. Also, when we run more experiments with different random seeds, we notice that this phenomenon heavily depends on the random seed.
>
> * We agree with the comments on $\ell_1$ and  $\ell_2$ losses. There are indeed other approximations to $\ell_1$ losses, such as Huber loss. However, approximations of this kind usually depend on a condition that requires extra hyperparameters, thus post greater challenges to the hyperparameter choices and optimizations. Therefore, even there might be other approximations, we consider that the cons outweigh the pros. On the other hand, we believe what conclusions we can draw formally given $\ell_2$ loss are still open mathematics/statistics questions. Therefore, while we agree that these questions are valuable, we tend to leave these questions for future study.
>
> * Thank you for pointing out the minor points. We will edit along with the suggestions.
>
> Thanks again for the overall positive assessment and constructive suggestions. Please let us know if there are more we need to respond.

---

### Official Review · AnonReviewer4 · 2020-10-28
**This paper generalizes the existing generalization results with specific assumptions into a framework, with the aim of pursuing invariance and robustness. However, the scope of robustness is restrictive and the assumptions seem not reasonable for me.**

**Rating:** 4
**Confidence:** 3

**Review:**

Strongness:
a) The paper unifies existing assumptions enforced to achieve robustness into a framework, by summarizing them into a set of general assumptions A1-A6. It can incorporate some types of transformations and models.
b) Among these generalization results, an unchanged term is the distance of representations between original samples and the ones after transformation. This is aligned with the conclusion in the literature.
c) By considering specific types of transformations (rotation, contrast, and texture), the experimental results show that the RA and RWA can improve a lot compared to VA and VWA, which validates the effectiveness of representation regularization (i.e., theorem 3.3 and lemma 3.4). Moreover, compared with existing methods specifically designed to boost the robustness with respect to the transformations considered, the methods show a significant improvement.
d) The paper is well organized and easy to understand.

Weakness:
a) My most concern lies in the validity of the assumptions, specifically A2, A5, and A6.
     i) For A2, consider the transformation set to be the l_infity additive noise for the adversarial attack, even the model specifically trained with robust (min-max) optimization, the label can still change. Besides, it seems that the framework cannot incorporate the transformations in the adversarial attack, which however is very important and cannot be ignored.
    ii) For A5, I cannot get the intuition of why equality holds. For any distribution, the "=" (rather than "\approx") seems too strong.
   iii) It seems that the purpose of A6 is for the convenience of the proof. Again, I cannot get the intuition of why the inequality holds especially when g(f(x; \theta)) \neq g(f(x'; \theta)), although I can understand that in this case, the transformation set is broad enough. But it is not sufficient to get this inequality since both sides are affected by the broadness of the transformation set.
b) As stated in the i) in a), this paper may not incorporate the transformations in the scenario of adversarial attack. Besides, the definition of the invariance is only restricted in the scope of considered transformation; correspondingly, the robustness of the paper only lies in the interpolation rather than extrapolation [1]. It is more meaningful to consider the invariance and robustness for general out-of-distribution settings, such as [1] and [2].
c) For experimental results, the Cross-domain performance on ImageNet-Sketch seems too low to have meanings. Again, the invariance and robustness should be discussed in a broader perspective. Besides, how's the improvement margin if the batch-normalization, default augmentation are implemented?
d) The theoretical analysis barely have some novel insights, since most of them are simple derivations of existing results.



[1] Krueger, David, et al. "Out-of-distribution generalization via risk extrapolation (rex)." arXiv preprint arXiv:2003.00688 (2020).
[2] Arjovsky, Martin, et al. "Invariant risk minimization." arXiv preprint arXiv:1907.02893 (2019).

---

> ### Author Response · Authors · 2020-11-13
> **Discussions with AnonReviewer4**
>
> [While we are still working on the questions raised by the reviewers, we hope to start the conversation with AnonReviewer4 early enough to discuss several points raised, for the reason that, with full respect, it seems AnonReviewer4 unfortunately misread some parts of the paper.]
>
> First of all, we would like to thank the AnonReviewer4 for the positive assessments of the strengths of our paper, such as the efforts to summarize existing works into one place, with empirical results well aligned to the literature and our derived understanding, and, most importantly, the empirical strength of our paper, because our paper is aimed to inspire more on the empirical side of the community.
>
> We would also like to thanks the reviewer for other constructive comments. However, we do feel like there is a need for us to further clarify some points. Therefore, we hope to have a polite and respectful conversation with AnonReviewer4 soon.
> * We can understand why the reviewer raises the concern that "the robustness of the paper only lies in the interpolation rather than extrapolation", however, our empirical results did test the extrapolation cases. For examples:
>     * Table 2 reports methods trained only considering 0-60 degrees rotations, tested with 0-60, 300-360 degrees rotations.
>     * Tables 3 & 4 report methods trained with our predefined transformation scopes, while the test data are generated independently, with no direct (mathematical) connections to our transformation scopes.
>     * The "Beyond" columns of Tables 7-12 (six tables in total) are testing the cases when samples are generated with other augmentations within the same augmentation scope.
>     * In case by "extrapolation" the reviewer meant extrapolation beyond transformation scopes: please notice that Table 3 & 4 are this in case, and please refer to our next point.
>
> * Why we believe a predefined transformation scope is necessary (even for the human)?
>     * For our discussion to go smoothly, please imagine a friend of us traveled back from Mars and brought back two creatures with arbitrary appearances (e.g., one with a dog ear and cat paw, the other with a cat ear and dog paw), she named them a cat and a dog, and asked us to tell which is which. Without a rough understanding of her naming preferences, there is no way to guarantee the correct classification.
>     * For the same reason, we do not think there can exist any generalization guarantees if we do not have a predefined transformation scope. In fact, even within the same transformation scope, without further assumptions, the generalization results over transformations not considered during training can be intricate to anticipate. e.g., "Beyond" columns of Tables 7-12 do not lead us to a clear conclusion.
>     * While we are happy to discuss the two papers the reviewer provides about extrapolation, the first paper tests extrapolation empirically, as we did in Tables 3 & 4. The second paper has relevant discussions but conclusions must be drawn with their own assumptions to regulate the shift of distributions.
>
> * We are grateful for the constructive comment "this paper may not incorporate the transformations in the scenario of adversarial attack", but we would like to remind the reviewer:
>     * Although we cite relevant papers, our paper is not about the adversarial attack.
>     * Even so, we tend to disagree with this remark. In Section C of our appendix, we showed that although it's hard for A2 to hold for any model, it holds for the regularized adversarial training case, where A2 is used. Although this paper is more about a unified view instead of a specific focus of adversarial attack, we are fairly confident our results are likely to hold for the adversarial attack. In fact, a more narrowly-focused method has been explored empirically in adversarial attack regime, see: Adversarial Logit Pairing (https://arxiv.org/abs/1803.06373).
>
> * We acknowledge that a theoretician may not feel impressed by our derivations. In fact, we do not recall that our paper has made such claims (if we have overclaimed, please let us know). Our goal of this paper is to show that related statistical discussions can lead to a unified view and further lead to a strong empirical method. Thus, we tend to focus on the empirical side of our problem: i.e., although we have strong assumptions, we have shown that our assumptions hold for a large number of samples in practice (Section A right after the introduction of assumptions, and Section C). Also, to the best of our knowledge, relaxing A5 to derive the generalization bound (without introducing other assumptions) requires non-trivial efforts, and this alone is still a challenging open-end research question.
>
> * ImageNet-Sketch seems a challenging dataset, and we followed the established procedures to compete with SOTA.
>
> We hope these discussions can help AnonReviewer4 rethink some unnecessary negative opinions of our paper. We are looking forward to further discussion.

---

> > ### Comment · AnonReviewer4 · 2020-11-25
> > **Response to the Author**
> >
> > Thanks for your response. I agree with your points about assumption A.2 on adversarial attacks, as validated in Tab.5. However, I'm still not convinced about the generalization ability on extrapolations, as you claimed. The reasons are listed as follows:
> >
> > - It has been discussed in [1,2] that optimizing over predefined augmented types is equivalent to robust optimization, which only considers the convex hull of considered transformations (i.e., $c_1 * T_1 + ... + c_n * T_n$, $c_1+...+c_n=1$, $c_i \geq 0$ for $i=1,...,n$). So far, I cannot get the intuition of why this method (which also leverage predefined transformations during training).
> >
> > - Besides, I'm not sure if the results in Tab.2,3,4 can be provided as evidences for the extrapolation generalization ability. Tab.2 only considers the 0-$\pi/3$, and the reverse direction of rotation, i.e., $-\pi/3$-0. It would be more interesting to consider the experiments with rotation angles from 60-180. The results in Tab.3,4 are too low to tell some insights from them; besides, it is this result that validates the failure of your model on extrapolation setting.
> >
> > - The predefined transformation is not a must. There is an increasing effort to leverage causal invariance to improve robustness. The key is to look for the "causal factors" of the prediction, which is invariant and unchanged under any circumstances. The "dog-ear, cat-pow and dog-pow, cat-ear example you make cannot prove the necessity of additional transformations; rather, it can only prove that we should define what is "cat" and what is "dog". Once it is well defined, we can leverage causal learning for more robust prediction.
> >
> > I'm looking forward to your answer and totally open to upgrading my scores.

---

> > > ### Author Response · Authors · 2020-11-25
> > > **Response to AnonReviewer4**
> > >
> > > Thank you for the response and the openness to upgrading the evaluation. We are glad that it seems the only disagreement is about the generalization ability on extrapolations. Below, we will respond point to point.
> > >
> > > * Unfortunately, the first point made by the reviewer is not complete. We are not sure what [1, 2] refer to, and we are not sure what the last sentence is asking for. Here is our answer based on our best guess of what the question is: even though there are previous works talking about the connection between predefined augmented types and robust optimization, one of our major arguments is the effectiveness of the method in terms of invariance. To the best of our knowledge (we cannot guarantee because what [1,2] refer to are missing), there are no works linking the augmentation, consistency loss, and invariance to the depth of our contribution.
> > >
> > > * Thank you for raising these concerns, here are our responses:
> > >   -  Just in case we were not clear before, in Table 2 where the models are tested by $[-\pi/3, \pi/3 ]$, the models are only trained with $[0, \pi/3 ]$, the interpolation is about $[-\pi/3, 0]$.
> > >   - Regardless, we followed the reviewer's suggestion to test other rotation results (detailed table at the very end of this response). The performances are telling the same story and favoring the regularized methods.
> > >   - Notice that, as the rotation degree increases, the performances for all methods, in general, will drop. This is inevitable (see the very last bullet point for the explanation).
> > >   - With full respect, we consider the concern raised as "The results in Tab.3,4 are too low to tell some insights" very subjective. Objectively speaking, these results are on par with SOTA. We're grateful that the reviewer is evaluating our paper's potential to be a perfect solution, while we will also appreciate the reviewer to build upon current advances of relevant methods.
> > >
> > > * We agree that the key is to look for "causal factors", which, in our opinion, is exactly the same as defining a transformation scope. The equivalence can be drawn when the transformation scope is "to perturb anything that is not the causal factors". Thus, transformation scope is exactly the opposite definition of "causal factors", and vice versa. Therefore, if one is a must, the other will be.
> > >   - For example, unfortunately, the experiment setup proposed by the reviewer to test against rotations with more degrees can be controversial. The increment of degrees gradually increases the transformation scope but eventually violates the "causal factors": a digit 6 will be a 9 after rotating 180 degrees, and the performances will guarantee to drop. This is only an example to show that transformation scope needs to be designed with the consideration of "causal factors", to support our equivalence argument above.
> > >
> > >
> > > | Rotation |    75   |    90   |   105   |   120   |   135   |   150   |   165   |   180   |   195   |   210   |   225   |   240   |   255   |   270   |   285   |
> > > |:--------:|:-------:|:-------:|:-------:|:-------:|:-------:|:-------:|:-------:|:-------:|:-------:|:-------:|:-------:|:-------:|:-------:|:-------:|:-------:|
> > > |   Base   | 0.3625  | 0.3463  | 0.3232  | 0.3045  | 0.2909  | 0.2815  | 0.2789  | 0.2793  | 0.2769  | 0.2720  | 0.2668  | 0.2618  | 0.2585  | 0.2581  | 0.2560  |
> > > |    ST    | 0.3881  | 0.3714  | 0.3508  | 0.3332  | 0.3204  | 0.3131  | 0.3115  | 0.3128  | 0.3109  | 0.3062  | 0.3000  | 0.2940  | 0.2905  | 0.2900  | 0.2880  |
> > > |    GC    | 0.2769  | 0.2595  | 0.2448  | 0.2325  | 0.2222  | 0.2156  | 0.2131  | 0.2137  | 0.2132  | 0.2106  | 0.2064  | 0.2024  | 0.1997  | 0.1986  | 0.1973  |
> > > |    ETN   |  0.3177 |  0.3044 |  0.2698 |  0.2677 |  0.2655 |  0.2813 |  0.3091 |  0.3295 |  0.2985 |  0.2803 |  0.2687 |  0.2558 |  0.2663 |  0.3202 |  0.335  |
> > > |    VA    | 0.6824  | 0.6409  | 0.5902  | 0.5474  | 0.5161  | 0.4928  | 0.4737  | 0.4609  | 0.4461  | 0.4326  | 0.4233  | 0.4174  | 0.4110  | 0.4044  | 0.3963  |
> > > |    RA    | 0.7314  | 0.6735  | 0.6185  | 0.5752  | 0.5439  | 0.5183  | 0.4974  | 0.4829  | 0.4684  | 0.4550  | 0.4452  | 0.4382  | 0.4308  | 0.4236  | 0.4164  |
> > > |    VWA   | 0.8239  | 0.7771  | 0.7159  | 0.6661  | 0.6256  | 0.5916  | 0.5630  | 0.5434  | 0.5287  | 0.5159  | 0.5043  | 0.4937  | 0.4833  | 0.4722  | 0.4624  |
> > > |    RWA   | 0.8229  | 0.7536  | 0.6968  | 0.6518  | 0.6152  | 0.5851  | 0.5602  | 0.5431  | 0.5279  | 0.5151  | 0.5038  | 0.4940  | 0.4841  | 0.4753  | 0.4684  |

---

### Official Review · AnonReviewer3 · 2020-10-28
**Weak Accept**

**Rating:** 6
**Confidence:** 4

**Review:**

##########################################################################


Summary:
The paper seeks to find how to train with augmentation to train robust and invariant models.
For this purpose, they evaluate if  models trained with augmentation are generalizable and their associated regularization techniques.

##########################################################################



Pros:
1. The paper tackles an important issue of training well with augmentation.
1. The theoretical analysis is strong and interesting.
2. The experiments are thorough and extensive.
##########################################################################


Cons:
1. While a number of related work is discussed in the Related Work section, it has not been considered in the experiments section. The authors should compare to the methods, for example cosine similarity hasn't been compared to.

2. The paper is not well written and hard to read. It mentions constant references to the appendix in the theory, without being complete in itself. It is hard to separate the paper's proposed techniques from baseline models, for example, VWA and RA are well known methods in the literature.

3. Key experimental details are missing: how many seeds were the experiments run for? It misses standard deviation results as well.

4. Have the authors compared to constrastive learning methods, for example,
A Simple Framework for Contrastive Learning of Visual Representations
Ting Chen, Simon Kornblith, Mohammad Norouzi, Geoffrey Hinton
##########################################################################


Rebuttal Questions:
Please address the cons above.
##########################################################################

---

> ### Author Response · Authors · 2020-11-17
> **Discussion with AnonReviewer3**
>
> Dear AnonReviewer3
>
> Thank you for the overall positive assessment. We appreciate your recognition of the significance of the topic, the strength of the analysis, as well as the extensiveness of the experiments. We are also grateful for the constructive comments raised as cons. Fortunately, we believe all of these points can be addressed. Please find below a point-to-point response to the questions raised.
>
> 1. Following the reviewer's suggestion, we continued to run and compare the cosine similarity method, the performance shows a comparable accuracy and robustness (worst-case accuracy) in comparison to $\ell_2$ method, but a far worse invariance score than $\ell_2$ method. For example, for the MNIST experiment, on average, cosine similarity shows an invariance score 0.2 less than $\ell_2$ method in rotation and contrast settings (Rotation: 0.4138 vs. 0.6540; Contrast: 0.2631 vs. 0.4128). We will add detailed comparisons to tables in Appendix E.
>
> 2. Thank you for this constructive comment, we will be able to bring the assumptions back to the main paper once we are allowed additional pages.
>
> 3. Thank you for the suggestions, we have run the experiments with more random seeds, and calculate the standard deviations. We notice that the results are fairly stable, as most of the standard deviations are on the magnitude of 1e-4. For example, below are the new tables for MNIST experiments:
>
> For Texture
>
> |`          Methods        `|`        Clean        `  |`        Worst         `|`        Invariance        `|
> | :----:| :----: |:----:| :----: |
> |Base	|0.9920$\pm$0.0004    |	0.9833$\pm$0.0031    |	0.9236$\pm$0.0004    |
> |VA	|0.9923$\pm$0.0004    |0.9902$\pm$0.0004    |0.9916$\pm$0.0045    |
> |RA	|0.9908$\pm$0.0001    |0.9905$\pm$0.0001    |1.0000$\pm$0.0000    |
> |VWA	|0.9919$\pm$0.0003    |	0.9900$\pm$0.0005    |0.9976$\pm$0.0031   |
> |RWA	|0.9911$\pm$0.0003    |	0.9909$\pm$0.0002    |1.0000$\pm$0.0000   |
>
> For Rotation
>
> |`          Methods        `|`        Clean        `  |`        Worst         `|`        Invariance        `|
> | :----:| :----: |:----:| :----: |
> |Base|0.9920$\pm$0.0004|	0.2890$\pm$0.0060|	0.2059$\pm$0.0040|
> |VA|0.9899$\pm$0.0013|	0.9364$\pm$0.0025|	0.5832$\pm$0.0222|
> |RA|0.9930$\pm$0.0003|	0.9526$\pm$0.0016|	0.6540$\pm$0.0434|
> |VWA|0.9466$\pm$0.0004|	0.9403$\pm$0.0008|	0.6427$\pm$0.0124|
> |RWA|0.9935$\pm$0.0004|	0.9882$\pm$0.0001|	0.9293$\pm$0.0379|
>
> For Contrast
>
> |`          Methods        `|`        Clean        `  |`        Worst         `|`        Invariance        `|
> | :----:| :----: |:----:| :----: |
> |Base|0.9920$\pm$0.0004|	0.2595$\pm$0.0100|	0.2074$\pm$0.0110|
> |VA|0.9890$\pm$0.0027|	0.9543$\pm$0.0261|	0.3752$\pm$0.0694|
> |RA|0.9937$\pm$0.0005|	0.9738$\pm$0.0074|	0.4128$\pm$0.0047|
> |VWA|0.8400$\pm$0.2633|	0.8080$\pm$0.3126|	0.3782$\pm$0.1362|
> |RWA|0.9938$\pm$0.0003|	0.9893$\pm$0.0003|	0.8894$\pm$0.0104|
>
> 4. Thank you for the suggestions of contrastive learning methods, while we are happy to expand our discussion to include the relevant papers, we notice the difficulties in experimenting with the method. For example, in the paper suggested, experiments are performed with a batch size of 8192 (Section 2.2). We also notice other papers suggesting that the success of contrastive learning methods depends on this large batch size: Figure 4(b) of Supervised Contrastive Learning (https://arxiv.org/pdf/2004.11362.pdf) compares the performance of a batch size of 6144 to a batch size of 500. However, we will still add relevant discussions to remind the readers of a potential alternative solution.
>
> Thanks again for the overall positive assessment and constructive suggestions. Please let us know if there are more we need to respond.

---

### Decision · Program_Chairs · 2021-01-07
**Final Decision**

**Decision:**

Reject

**Comment:**

The reviewers brought up many important concerns about this paper. On the positive side, the understanding of data augmentation is an important topic in deep learning, having good theoretical results is interesting here , and the experiments seem to do an okay job of backing up the theory. On the negative side, presentational issues make the paper difficult to follow and mischaracterize the results. A major issue is that some of the assumptions are hidden in the appendix and are not stated formally, and other assumptions are stated in a much weaker form, then made suddenly stronger when the theorems are stated. For example, Assumption 2 as stated holds trivially for any dataset as long as the possible data-augmented versions any two different examples are disjoint (just choose the discrete metric on the images of the examples under the data-augmentation function); however, in every theorem that uses A2, the distance chosen is restricted to be the L1 norm. Other Assumptions are stated strangely: for example, A1 says "i.e., for any $a_1(), a_2() \in A$, $a_1(x_1) ⫫ a_1(x_2)$ for any $x_1, x_2 \in X$ that $x_1 \ne x_2$. But what is the point of introducing $a_2$ if it's never used in the formula? And what is the meaning of the symbol ⫫? Normally, this is used for conditional independence, but there aren't any random variables in this formula ($a_1$ and $a_2$ are defined as just functions, not random functions, and $x_1$ and $x_2$ are just examples and aren't random variables either). This paper will be much stronger with these presentational issues cleared up.